# The Drivers of Change for Future Learning: How Teachers Were Taught in the COVID-19 Crisis and What Will Come Next?

Usama M. Ibrahem [1,2,*] , Hussein M. Abdelfatah [2], Dalia M. Kedwany [3,4], Abdullah Z. AlMankory [3], Ibrahem M. Diab [3,5] and Rabab A. Abdul Kader [1]

1 Collage of Applied, University of Ha'il, Hail 2240, Saudi Arabia
2 Ismailia Education Collage, Suez Canal University, Ismailia 41522, Egypt; hussein.abdelfatah@edu.suez.edu.eg
3 Collage of Education, University of Ha'il, Hail 2240, Saudi Arabia; az.almankoury2@uoh.edu.sa (A.Z.A.); i_dyiab1@uoh.edu.sa (I.M.D.)
4 Kindergarten College, Alexandria University, Alexandria 21526, Egypt
5 Faculty of Physical Education, Sadat City University, Sadat 32897, Egypt
* Correspondence: u.abdelsalam@uoh.edu.sa; Tel.: +966-567-578-899

**Abstract:** The COVID-19 pandemic wreaked havoc on education and resulted in huge changes. This research paper investigates on the factors driving change for future learning by studying the training of teachers during the COVID-19 crisis and their perceptions regarding the future of teaching. The study employed the methodology of opportunistic or emergent sampling to collect data from teachers aged 30–50 years who had experience in conducting online classes in different schools in KSA. The study suggests that effective virtual education depends on careful instructional design, audience consideration, and a systematic implementation model that produces various forms of teaching tailored to educational objectives. The research findings can guide future decisions about implementing online teaching, and the dimensions identified in this research can be compared with previous studies to derive key learning axes for future schools. The pandemic transition presents an opportunity to develop sustainable paradigms for future generations.

**Keywords:** future pedagogy; COVID-19; online teaching; future learning; emergency remote teaching; learning crisis; future educational needs

## 1. Introduction

COVID-19 has had the greatest impact on education systems ever. It affected about 1.6 billion students in 200 nations, which altered their entire lives. Social isolation and movement restrictions have disrupted education. Students, parents, and educators worldwide have felt the unanticipated rippling impact of the COVID-19 outbreak as schools have closed. Many home-living students endured psychological and emotional distress and could not interact successfully.

During COVID-19, nations and educational systems that performed well made sure that their teachers could use technology. Teachers must be proficient in using technology as well; it is not enough to simply provide it. Different teachers will use technology in different ways [1,2]. Diverse tactics will be required to support them. Technology should be incorporated into everyday requirements like HR systems as well as teaching and learning activities as school systems mature [3]. According to Kapasia et al. (2020) and Siddiqui et al. (2023), tailored interventions can help students have a productive environment in which to learn. Strategies are desperately needed to create a resilient educational system that will ensure the development of young brains' productivity and employable skills [4,5].

The educational community came to the realization that, throughout the pandemic, teachers and students were driven to adapt online teaching and learning platforms to meet their educational demands. Some instructors and students were already proficient with

social media platforms like Facebook, Twitter, WhatsApp, and Instagram. The ease of use of online learning tools like Zoom, Cisco WebEx, Google Meet, etc. is a sign of effective learning transfer. Also, there are a number of beneficial educational programs that can be downloaded for free and are simple to use, like Office 365, Google Classroom, and many others [5,6].

Teachers, as the frontline in implementing online learning, are indicated to have taken various actions related to effective learning techniques and procedures during the pandemic [7,8]. Teachers play the most important role in implementing the changes from direct learning in the classroom to online learning because they are the controllers in the learning process [9]. Losses, gains, and transformations during the pandemic were a result of the shifts in teachers' work during COVID-19. Yet, many analysts also anticipate that some of these changes will endure after the pandemic ends. Some of this is motivated by the knowledge that climate change, pandemics, and other natural disasters may disrupt regular schooling more frequently in the future, necessitating the creation of more adaptable and responsive systems of teaching and learning now in preparation for those eventualities [3,10].

As a timely gap mechanism for meeting the demands of the COVID-19 pandemic, online teaching has been implemented as a viable alternative, well within its limits and limitations. It is a moot question whether it can be effective in the future or not. There is a need for understanding activities and strategies to be undertaken post-pandemic to ensure the continuity of online learning [11]. It is important to emphasize that these are emerging pedagogical trends. We need more experience, evaluation, and research to figure out which ones will have lasting value and change the system for good. Time must be spent innovating and implementing alternative educational systems and assessment strategies. The COVID-19 pandemic has provided us with an opportunity to pave the way for introducing digital learning [12].

Preparing for post-COVID-19 is planning for an unexpected future regarding health, education, governance, investment, trade, and electronic infrastructure. To balance these anxieties and tensions during a crisis, technology must be studied thoroughly. Digital proficiency requires self-reflection and critical thinking in high-risk areas for students and teachers. The article will explore how the essential elements of education and its organizational culture have changed and may change in the future. Changes in the emotional dynamics of teaching and learning, changes in the professional capital of teachers—particularly the social capital—and changes in their professional, particularly pedagogical, expertise are all related to changes in students' learning outcomes.

## 2. Research Questions

1. What are perceptions of teachers of online teaching during the COVID-19 pandemic?
2. What are teachers' perceptions of teaching in the future?

## 3. Literature Review

During COVID-19, there was a shift from traditional pedagogic and pedagogical models to heutagogy and a cybergogy approach. Heutagogy is a student-centered learning strategy where learner-determined and self-directed learning occurs. Schools across the country have grappled with similar challenges in response to COVID-19 [6,7]. There could be real benefit from creating opportunities for principals and schools to share their experiences and innovations and be informed by other approaches and evidence of best practices [13].

### 3.1. Crisis Pedagogy Online

The transition from traditional face-to-face to online learning can be a completely different experience for both learners and instructors. Online learning is characterized as learning experiences in synchronous or asynchronous situations using various devices with an internet connection (e.g., mobile phones, laptops, etc.) [12]. Online education

pedagogy may depend on educators' and students' ICT experience. E-pedagogy is a branch of pedagogy. Therefore, its content should represent didactics and education [14]. Teachers can design educational courses, training, and skill development programs on unified communication and collaboration platforms like Microsoft Teams, Google Classroom, Canvas, and Blackboard [2,9].

The flipped classroom is a simple strategy for providing learning resources such as articles, pre-recorded videos, and YouTube links before the class. The online classroom time is then used to deepen understanding through discussion with faculty and peers [11,15]. A teacher conducts online instructional activities based on learning outcomes and each group's learning [13,14]. Collaborative approaches to knowledge construction/building communities of inquiry and practice emphasize allowing students to construct knowledge through questioning, discussion, sharing perspectives and sources, analyzing resources from multiple sources, and receiving instructor feedback. Furthermore, social media encourages the development of communities of practice in which students share their experiences, discuss theories and problems, and learn from one another [8].

Technology is capable of identifying trends and conditions that may require intervention. Instructors can improve their ability to support students significantly. This is referred to as "data-driven learning" by some instructors. Yet, it is people-driven learning [12]. Several of these new educational approaches may have an impact on the future education system. Caring for educators is an important aspect of recovery and a long-term education paradigm [9,16].

### 3.2. Educational Crisis Effect

The crisis encouraged innovation and improvement within the education sector in order to ensure the continuity of students' learning. People's attention has also been attracted to longer-term prospects as a result of the specific developments that occurred during the pandemic. These include possible increases in home-schooling rates; increased use of virtual and digital learning within in-person school environments; shifts to hybrid patterns of curriculum delivery; increased use of outdoor learning environments both on and off the school campus; and new, virtual forms of professional collaboration with educators in other schools and/or at more convenient and/or flexible times in relation to teachers' schools too [17]. Overall, teachers have a positive perception of online education during the COVID-19 pandemic and young teachers have more actively engaged in online learning. Further, e-learning not only enhanced teachers' knowledge but also improved their technical skills [4,18]. Online learning allows students to do self-study under the guidance of the course lecturer, and that provides them a level of convenience not allowed under the traditional teaching methodology [11,19].

### 3.3. Teachers as Educational Heroes of the COVID-19 Pandemic

Technology and infrastructure are crucial, but the most important aspect of effective education is qualified teachers. The teachers' creativity and hard work prevented this generation of youth from falling behind. Remote education was quite successful online, using multiple platforms [20]. The teachers stayed enthusiastic and determined in the face of numerous hurdles. Regardless of how much technology our instructors possessed, they all found ways to work around it and were happy to share creative solutions or free resources they had discovered [19]. They also wanted to gain knowledge from their international colleagues. Some had witnessed favorable advances, particularly in adopting technology where there had previously been mistrust, with some students adjusting exceptionally well to the online world [15].

During the COVID-19 pandemic, teachers play a key role in making sure that education keeps going, despite the problems caused by the crisis. The teachers have had to learn how to use new ways to teach, like online classes and distance learning, to make sure that their students continue to receive a quality education [21]. They have also had to give students who are struggling with dealing with the effects of the pandemic emotional

support and guidance. The teachers have become heroes in many ways during this crisis because they have gone above and beyond their normal duties to ensure their students do not fall behind. In these hard times, it is important to recognize and appreciate the hard work and dedication of our teachers [18]. Quality and equity in education systems can only be achieved by providing teachers with appropriate tools, the time to train and collaborate, and mutual trust between systems and people.

### 3.4. The Transition from a Traditional Expository to a Hybrid Model

Educational stakeholders need support in developing transformed instructional models and encouraging teachers to learn new educational practices for the future. Using technology like artificial intelligence (AI), big data, learning analytics, and a range of new devices and tools to rethink education will help transform the roles and relationships of students, teachers, and parents [16,18,20] as follows:

- **Students will learn at their own pace in flexible, often collaborative methods inside and outside classrooms. They can pursue their interests and be challenged if needed**.
- The teachers **will have access to real-time, individualized data on their students' academic and emotional progress, allowing them to create new challenges and provide the right support for each child's advancement**.
- Parents will be better connected to, and involved with, their child's education with certainty, detail, and confidence.
- Curriculum adaptation, flexibility, and contextualization should prioritize learning objectives and content that improve crisis knowledge and response, including care and health, critical and reflective thinking about information and news, and understanding social and economic trends.
- The classroom, as we have known it for centuries, will also be reimagined. It is predicted that technology will see schools morphing into "learning hubs".

All this emphasizes the need for a pedagogical approach that relies heavily on the social and collaborative components of learning as a starting point for the development of online teaching and learning practices.

## 4. Data Collection

The purpose of this study is to explore issues and concerns about the introduction of online teaching during COVID-19. We have utilized the methodology of opportunistic or emergent sampling. This is considered to be a flexible research and sampling design that allows us to carry out qualitative research when we are still exploring.

The participants in the study provided their written responses to a "standardized open interview", which comprised a unified sequence of pre-prepared, consecutive questions. The responses were used to collect data for the study. It is beneficial to ask all of the participants the questions in a methodical order since this has the advantage of reducing the influence of subjective assessments. The virtual interview is divided into two parts. In the first section, demographic information about the participants is collected, and in the second section, open-ended questions on the participants' perspectives on the future learning ecosystem in relation to teaching during the pandemic are posed.

### 4.1. Participants

The participants (as opportunistic or emergent sampling) were selected according to their online experiences in the field of teaching and conducting online classes. The study recruited teachers aged 30–50 years from different schools in KSA. The purpose of the study was explained to the participants.

### 4.2. Methodology

The study community represents teachers in Saudi schools, 264 teachers in 85 Saudi schools, as in Table 1.

**Table 1.** Demographic information for the study sample: school, academic rank, gender, and years of experience.

| No. | Demographic Information | | Number of | | % of Total |
|---|---|---|---|---|---|
| | | | Teachers | Schools | |
| 1 | School cities | Hail | 30 | 8 | 11.4 |
| | | Dammam | 20 | 5 | 7.6 |
| | | Gizan | 15 | 9 | 5.7 |
| | | Mecca | 35 | 8 | 13.3 |
| | | AL Madinah | 22 | 6 | 8.3 |
| | | Al Qussaim | 30 | 8 | 11.4 |
| | | Al Riyadh | 24 | 7 | 9.1 |
| | | Jeddah | 22 | 5 | 8.3 |
| | | Al Jouf | 16 | 6 | 6 |
| | | Al Baha | 26 | 7 | 9.8 |
| | | Tabuk | 24 | 6 | 9.1 |
| 2 | Gender | Male | 122 | | 46.2 |
| | | Female | 142 | | 53.8 |
| 3 | Highest level of education | Bachelor's | 172 | | 65.2 |
| | | Higher Diploma | 38 | | 14.4 |
| | | Master's | 42 | | 15.9 |
| | | Ph.D | 12 | | 4.6 |
| 4 | Years of experience | Less than 5 years | 124 | | 47 |
| | | From 5 years to less than 10 years | 32 | | 12.1 |
| | | From 10 years to less than 15 years | 46 | | 17.4 |
| | | From 15 years to less than 20 years | 44 | | 16.7 |
| | | 20 years and over | 18 | | 6.8 |

*4.3. Data Collection*

Participants' written responses to the "standardized open interview" were used to obtain research data. The open interview comprises a set of pre-planned, consecutive questions that are asked to all participants in the same format and system. While attitudes and instantaneous flexibility are limited in this strategy, systematically asking all participants gives an advantage by decreasing the influence of interlocutor and subjective assessments. The interview form is divided into two sections. The first section regards demographic information on the participants. In the second half, open-ended questions about the participants' opinions on the future of distance learning in light of teaching during the COVID-19 pandemic are presented.

*4.4. Data Analysis*

MAXQDA 2022 analyzed transcribed interviews. Data collection and analysis were simultaneous. The qualitative data analysis software MAXQDA (version 2022) has become popular recently. The MAXQDA program performs many qualitative analysis and research tasks, including content analysis, data classification, and binary analysis, by focusing on the relationships between elements, cross-classifying and analyzing data, linking data, and searching for patterns and concepts. Textual analysis and thematic analysis, which involve comprehending texts' semantics and meanings and finding patterns [22,23], were carried out. Data were analyzed inductively. To avoid overlap, participants' replies, especially in

the first phase, were coded using keywords. MAXQDA has coded the data. Thematic maps indicate that concepts are organized by level and then developed into potential interactions. The analysis team then reviewed all codes and classifications and considered integrating them to simplify them. Inductive analysis identified participant responses to research questions. Study axes presented qualitative results.

After transferring all the relevant files, MAXQDA created a new project called "Drivers of Change for Future Learning" and 15 resource volumes in Microsoft Office Word format. Encoding involved linking extracts in pools (nodes). Creation of tree nodes followed. Thematic analysis reveals the primary effects of online teaching on future learning. Congruency was used by selecting expert viewpoints to review and criticize the research processes, assess its data, and reach outcomes similar to those achieved to ensure the validity of the qualitative data and achieve stability in qualitative research.

For the purposes of this study, at first, a semi-structured interview was started with one person, and the interview in the 264th interviewee reached saturation point. In the cases where the interviewees permitted, the data were recorded by audio, and in some cases, the data were recorded through note-taking. After each interview, the data were entered into MAXQDA Update 2022.4 software in Word 2 format and coding was done for 264 interviews. Next, to ensure the reliability of the measurement tool, the coding results were reported to the interviewees so that they could confirm whether the coding results were what they intended. Therefore, at this stage, some codes were deleted and others renamed.

## 5. Findings

To answer the research question, 264 teachers were interviewed about how they would feel about teaching online in the future based on what they had learned from teaching during crises. The results of semi-structured interviews during several stages of coding led to the identification of 264 repeated, basic codes, and finally, after merging the repeated codes, 13 open codes were extracted. In the second step, the core codes were identified and categorized into 59 core codes, the results of which are reported in Table 2 and Figure 1 below:

**Table 2.** Participants' perspectives on the future learning ecosystem.

| Optional Code | Axial Code | Themes |
|---|---|---|
| Future learning | Customized learning | Online and face-to-face instruction |
| | | Blended learning |
| | | E-books |
| | Creating an environment for content production as a team in schools through education and training | AR, VR |
| | | Learning analytics |
| | | Technical training |
| Instructional design, curriculum, and teaching strategies | Production of multimedia content in different formats | Team-based e-projects |
| | | Portfolios |
| | | E-tasks and assignments |
| | | Social media communication |
| | | Traditional and electronic teaching approaches must be mixed |
| | | Student-centered design |
| | | Communicative and cognitive theories |
| | | Electronic teaching methods |

**Table 2.** *Cont.*

| Optional Code | Axial Code | Themes |
|---|---|---|
| Staffs' future needs | Support for content training in case of learning problems | Provide educational material effectively |
| | | Technological training and strategies |
| | | Improving teachers' digital abilities |
| | | Professional development for teachers and virtual professional community |
| | | 21st century teacher skills |
| | | Ability to accept change |
| | | Motivating teachers |
| | | Technical, social, and ethical training |
| | | Support collaborative professional development |
| Support and technical support | Infrastructure and logistics | Improve the structure of e-learning |
| | | Improve the infrastructure and its technical and technological equipment in schools |
| | | Lack of internet/insufficient internet |
| | | Enhancing the responsiveness of educational platforms |
| | | Using interactive videos and intelligent software |
| | | Smart sensors |
| | | Improve the structure of e-learning |
| | Focus on sustainability | Using renewable energy sources |
| | | Reducing waste and promoting eco-friendly habits among students |
| | | Flexible spaces adapted to the needs of the learner |
| | | Adapting to environmental changes |
| *Technologies, resources, and activities* | Improve learning | Providing students with a wide selection of resources and activities |
| | | Interactive and interesting learning experiences |
| | | Enhance classroom learning |
| | | New educational technology such as learning management systems (LMSs), online collaboration tools, and mobile apps also improve learning results |
| | Invest in modern technology | Easy access to resources from anywhere |
| | | Use of smart devices and applications |
| | | Providing access to resources such as books, technology, and educational materials |
| | | Online portals for parents |
| | | Easily manage class schedules and communicate with students |
| *Future learner evaluation* | Adopt alternate evaluation | Portiflio, e-projects, and question banks |
| | | Peer evaluation |
| | | Self-evaluation |
| | | Generating deeper and more authentic e-assessment strategies and methods |
| | Focusing on formative evaluation | Employing different types of e-evaluation effectively (diagnostic–formative–final) |

**Table 2.** *Cont.*

| Optional Code | Axial Code | Themes |
|---|---|---|
| Future teachers' assessment | Digital skills | Learning analytics |
| | | Teacher proficiency in using digital skills in teaching |
| | | Use of social media applications |
| | | Interaction with and response to recent digital developments |
| | Advanced training courses in specialties | Classroom observations |
| | | Advanced training courses in specialties |
| | | Cooperation with subordinates and colleagues |
| Parental influence on future learning | Strengthening the connection between the school and the family | Encourage and guide their children as they navigate the online learning environment |
| | | Monitor their children's progress and communicate regularly with teachers |
| | | Ensuring children's access to educational resources through e-learning platforms |
| | | Contribute to a calm learning environment at home |
| | | Provide support and guidance to their children in navigating the online learning environment |

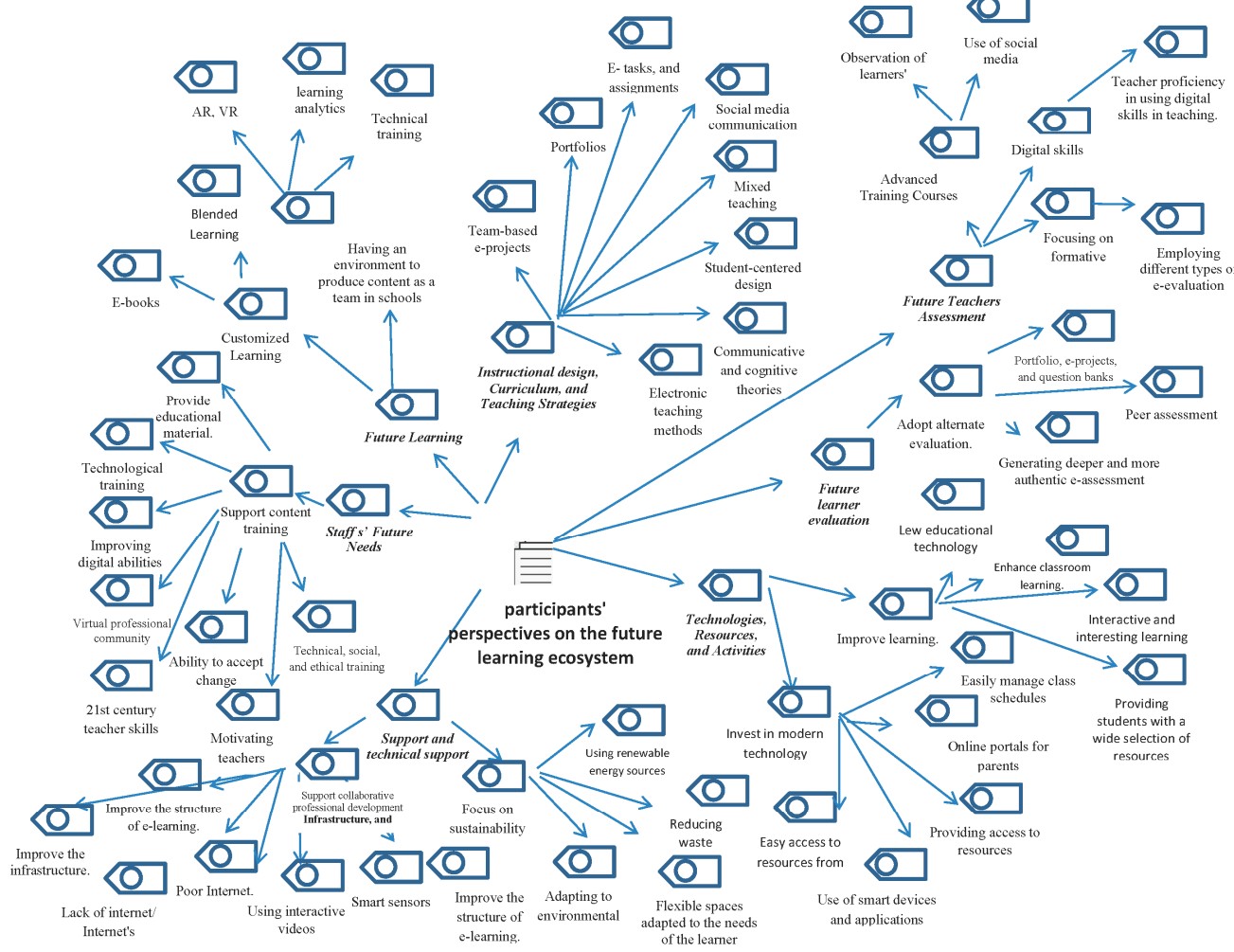

**Figure 1.** Future perceptions as represented by MAXQDA Update April 2022.

The previously mentioned themes were among the views expressed by the sample, as shown below.

### 5.1. Future Learning

In exceptional circumstances, e-learning (hyper learning) has become a strategically indispensable option. Jayanthi et al. (2023), Marwaha (2021), and Theiri and Alareeni's (2023) investigation findings were confirmed [24–26]. The following are the teachers' perspectives.

"Online and face-to-face instruction make education more flexible" (Pos. 31; 264, 111).

"In order to increase diversity, efficacy, and interactivity, educational institutions constantly integrate e-learning technology with face-to-face teaching" (Pos. 43; 104; 106; 200).

"Hybrid education encourages the effective use of different teaching methods and helps students improve their skills" (Pos. 21; 25; 215).

"Turning all books into interactive e-books rich in multimedia and depending on iPads or smart devices with learners" (Pos. 82; 99; 141; 122; 64).

Many teachers believe that they need further technical training, especially in blended learning, AR, VR, learning analytics, and interactive e-learning. Online learners perform slightly better than face-to-face students [27,28]. LMS investments increased due to COVID-19. AR apps, blended learning, and e-evaluation include curriculum revision, instructional modalities, and school–home assignments [25,29].

Results show teachers' awareness of technological development. Technology-supported capabilities offer a greater range of guiding methods, maximizing the benefits of each location and proving that one-on-one instruction is best.

### 5.2. Instructional Design, Curriculum, and Teaching Strategies

Many teachers modified performance measurement during the pandemic. They changed evaluation methodologies. Changes included student-led instruction, team-based e-projects, portfolios, flexible deadlines for activities, tasks, and assignments, and social media communication. We can describe the schools' critical points as follows:

"Classrooms no longer accept traditional teaching approaches (Pos. 61; 85; 150). To meet learners' requirements and tendencies, traditional and electronic teaching approaches must be mixed" (Pos. 20; 142; 221; 261).

"Online learning requires student-centered design. Content alone produces terrible multimedia" (Pos. 28; 26; 89; 222).

"Prioritize equality and student worry (Pos. 6; 20; 117; 145). Instructors should not overuse simultaneous video conferencing to accommodate kids with weak internet access or other families that need bandwidth" (Pos. 9, 123, 152).

"Due to changes in the learning environment and its diversity between online and face-to-face, teachers must retrain on communicative and cognitive theories" (Pos. 19; 29; 117).

"My teaching priorities changed. Instead of wondering how to teach, I've decided to shift my classes to a distant learning environment using proper electronic teaching methods" (Pos. 5; 16; 155; 156; 235).

The crisis forced educational sector stakeholders to rethink the current teaching strategies and curriculum. Using a learner-centered approach, virtual learning provides adaptable learning [15,28]. The future of schools will depend on their ability to provide open and adaptive models for the diversification of teaching, learning, and evaluation [1,30].

"Prioritize equality and decrease of student fear. Teachers should not rely disproportionately on simultaneous video conferencing to include students with inadequate internet infrastructure or other families in need of internet capacity" (Pos. 14; 99; 149; 246).

"Online education necessitates a student-centered design (Pos. 30; 120; 148; 225). Focusing solely on content will develop more effective multimedia" (Pos. 39, 101; 102; 231).

"My instructional priorities have changed (Pos. 15; 46; 107; 205). Instead of worrying how to transfer my knowledge, I've opted to adapt my classes to a distant learning setting using proper electronic teaching methods" (Pos. 5; 57; 126; 214).

Existing curriculums for remote instruction have not been established and adapted effectively. Hence, when remote instruction is required, an alternative model must be employed. Instructors feel that emergency remote teaching permits a pedagogical transition to a less structured and more enjoyable method of instruction [24]. Humanizing education can be operationalized by transcending cognitive-only techniques and becoming reflexive [31]. It is envisaged that teachers would employ pedagogical innovations to improve student participation through empathy and caring [32]. Schools and educators must be able to adapt and align their courses to meet the changing societal and individual learning demands. A curriculum of the future should allow pupils to acquire new talents for the future [17].

### 5.3. Staffs' Future Needs

The global pandemic revealed a large deficit in teacher preparation, which is essential for emergency remote instruction [20,33,34]. The majority of teachers agreed with the significance of technological training and strategies, and they stated the following:

"The teacher's status has not changed significantly as a result of the epidemic, but the challenges have increased. They must be able to overcome obstacles and provide educational material effectively" (Pos. 22; 58; 88; 139).

"Improving teachers' digital abilities leads to greater performance, easier communication with students, and the development of course presentation methods" (Pos. 98; 110; 230; 249; 260).

"Professional development for teachers and virtual professional groups enable teachers to smoothly clarify curriculum knowledge, facilitate materials management, and teach more creativity" (Pos. 38; 192; 211; 237; 262).

"The emergence of distant training requires a reconsideration of teacher training in to understand its future function" (Pos. 49; 100; 128; 171).

"A knowledge with a 21st Century teacher's skill set, an emphasis on LMS applications, and familiarity with modern technology tools are required technological abilities" (Pos. 53; 77; 135; 202; 254).

"Teachers must use technology to teach after COVID-19. Improving his subject presentation applications is crucial" (Pos. 10; 72; 136; 165; 167).

"A knowledge with a 21st Century teacher's skills and experience, an emphasis on LMS applications, and familiarity with modern technology tools are required technological abilities" (Pos. 53; 161; 238).

New pedagogical training requirements for teachers are linked to developments in educational process organization approaches [21,35]. One of the most important requirements for successfully integrating technology is the ability to accept change. Preparation and motivation of teachers are essential for the successful integration of technology in classrooms [36,37]. Before ICT can be utilized effectively, teachers need technical, social, and ethical training and assistance in ICT and pedagogy [38]. Also, it would be crucial to support collaborative types of professional development among teachers, such as teachers' networks, as this would enable them to learn from their colleagues [3].

### 5.4. Infrastructure and Logistics

In the context of strengthening the technical infrastructure of schools to prepare for their future role, teachers' perceptions of the significance of this factor were as follows:

"There is a need to improve the structure of e-learning; the need to improve the infrastructure and its technical and technological equipment in schools" (Pos. 1; 82; 97; 175; 189).

"Future improvements must be made to the Internet's deficiency, which impedes communication with teachers, and to the technological challenges (poor Internet, enhancing the responsiveness of educational platforms" (Pos. 87; 113; 134; 195; 206).

"Using interactive videos and intelligent software. To inspire reflective thought and look for immediate or delayed feedback on the questions given" (Pos. 7; 36; 66; 125; 262).

"Lighting that uses less energy and automatically resizes for occupancy and weather. Smart sensors that track temperature and air quality to provide a healthy learning environment" (Pos. 100; 147; 198; 218).

"Focus on sustainability: Schools should adopt sustainable practices such as using renewable energy sources, reducing waste, and promoting eco-friendly habits among students" (Pos. 99; 243).

"The traditional classroom setup may not be suitable for all students. Schools should consider creating flexible spaces that can be adapted to different learning styles and needs" (Pos. 87; 251; 256).

The crisis demonstrated that a strong IT infrastructure is essential for future schooling. Adapting to environmental changes quickly takes preparation and technological resources [39]. The World Bank states that school systems are not ready to offer online learning to all students. Technology often outpaces decisionmakers' cost and infrastructural support [38]. Online and blended learning require ICT infrastructure, tools, hardware, and software. ICT's application in academic courses has grown exponentially. Hence, schools are adding LMS, AR, VR, IOT, and educational blogs to their pedagogy and practice [40,41]. Governments and education providers should promote the construction of educational information, equip teachers and students with standardized home-based teaching and learning equipment, conduct online teacher training, and support academic research into online education, especially education to help students with online learning difficulties [41].

Meta-synthesis of the literature shows that multimedia, e-learning, adaptive learning, semantic web, and ICT-enhanced content have become more popular in recent years to improve teaching and learning. All studies agree that smart technology, especially the Internet of Things, will be vital to all aspects of learning [25,38].

*5.5. Technologies, Resources, and Activities*

New educational technologies, e-resources, and e-activities improve learning by providing students with a wide selection of resources and activities. These technologies give students interactive and interesting learning experiences that help them understand complex concepts more easily [42,43]. Online textbooks, films, simulations, and virtual laboratories offer students a variety of knowledge to enhance classroom learning. These resources can be accessed anywhere and are typically more current than textbooks [24].

E-activities such as online quizzes, games, and interactive exercises help pupils comprehend crucial ideas. These exercises are frequently exciting and engaging, which keeps pupils engaged and interested [43]. New educational technology such as learning management systems (LMSs), online collaboration tools, and mobile apps also improve learning results. LMS platforms allow teachers to design online courses that students can access anywhere, anytime. Online collaboration solutions allow students to work together on projects regardless of location. Students may learn anywhere with mobile apps [42]. Teachers' perspectives resulted in the following views.

"Turning all books into interactive e-books rich in multimedia and depending on iPads or smart devices with learners" (Pos. 27; 58; 69; 127).

"Schools should invest in modern technology such as interactive whiteboards, tablets (Pos. 158; 170), and laptops to make learning more engaging and interactive" (Pos. 87; 169; 228).

"Providing access to resources such as books, technology, and educational materials is crucial for student success" (Pos. 34; 94; 115; 143).

"Online portals for parents to access information about their children's academic progress and school activities" (Pos. 30; 69; 79).

"Digital scheduling tools that allow teachers to easily manage their class schedules and communicate with students" (Pos. 67; 108; 137; 178).

The utilization of new educational technology, e-resources, and e-activities has transformed how students learn and teachers teach. These tools have made learning more dynamic, engaging, and individualized. They have affected learning outcomes in the following ways:

- Customized Learning: With the support of e-resources and e-activities, teachers may develop personalized learning experiences for each student based on their specific needs and skills. This allows students to learn at their own speed and style [44,45].
- Increased Engagement: Educational innovations such as gamification, simulations, and virtual reality have made learning more entertaining for students. They are more inclined to learn when they are having fun [41].
- Information Access: Online libraries and databases give students access to a wealth of knowledge. This broadens their knowledge and develops critical thinking [30].
- Collaborative Learning: Online conversations and group projects encourage student collaboration. This helps them build workplace-relevant collaborative skills [45].
- Flexibility: E-learning lets students study anywhere, anytime. Its flexibility benefits adult learners with work or family commitments [46].

In conclusion, the employment of modern educational technologies, e-resources, and e-activities has significantly improved learning outcomes by making learning more individualized, engaging, accessible, collaborative, and flexible.

*5.6. Future Learner Evaluation*

E-evaluation in the environment of distant education is one of the most crucial aspects of the educational process. This is because it is related to measuring learning outcomes, evaluating learners' educational task completion, and determining their levels. In this context, the perspectives of teachers revealed the following opinions.

"It is important to adopt alternate evaluation methods that can increase the effectiveness and validity of the evaluation process (Pos. 40; 79; 190), such as Portiflio, e-projects, and question banks" (Pos. 70; 78; 259).

"Focusing on formative evaluation due to its capacity to measure and evaluate the various skills of learners" (Pos. 34; 67; 210).

"Interest in online discussions and projects is increasing (Pos. 116; 142). Peer assessment, establishing technical and quality alternative assessment tools and processes, generating deeper and more authentic e-assessment strategies and methods to adjust learning outcomes" (Pos. 16; 48; 209).

"Employing different types of e-evaluation effectively (diagnostic-formative-final)" (Pos. 132; 180; 240).

"Variety of student evaluation instruments (research, presentations, performance tests, oral exams, and group projects)" (Pos. 18; 109; 140; 177).

Evaluation is an essential component of learning and teaching since it results in the achievement of learning outcomes by students. Because courseware systems will test students' abilities at each stage, assessing their competencies via Q&A may become obsolete or insufficient. Furthermore, many individuals believe that today's assessments are meant to encourage pupils to cram their content and forget about it the next day [5,47]. Conventional forms of evaluation are being chastised these days for leaving students with compressed knowledge for marks rather than the skills required for proficiency [36,44].

As a result, diverse assessment choices that can assist in employing online learning must take into account challenges in the educational environment [48,49]. There are numerous internet platforms available for evaluation and each has its own set of benefits and drawbacks. There is a need to use learning management system (LMS) elements that improve the reliability of online reviews [33,48]. Assessments can take the shape of virtual presentations, interaction models, oral presentations, creative projects involving 3D modeling and graphics, skits or plays, weblog journaling, one-on-one conferencing, and other activities [29,35]. This kind of assessment can be used to judge authenticity and performance and hence could be a relief measure in this time of rapid educational upheaval.

### 5.7. Future Teachers' Assessment

One of the topics in teachers' perspectives that might change how they teach in the future is how a teacher is evaluated in distance education. It improves learning outcomes and strengthens teacher practice [46,47]. The quality of the teacher evaluation process can ensure the quality of teacher performance if it is carried out appropriately [30,49]. According to teachers, the following criteria should be used to assess teachers' effectiveness in the future:

"The most important evaluation criteria are teacher proficiency in using digital skills in teaching (Pos. 166; 188), his use of social media applications, his access to advanced training courses in his specialty, his observation of learners' characteristics and individual differences (Pos. 185; 201; 204), his interaction with and response to recent digital developments, the student's semester and final results, and his cooperation with subordinates and colleagues (Pos. 202; 231; 245). The degree of his work performance and professional requirements, the implementation of the appropriate suggestions from specialized supervisors, the attendance of classes and daily work" (Pos. 131; 153; 258).

"The most significant way to evaluate a teacher is by assessing student performance and knowledge. Some think this is unfair to the teacher because some children have bad levels that cannot be improved (Pos. 39; 97; 127). Yet, in my opinion, these kids' forgetting has little effect, and the teacher must develop his pupils' levels using all conceivable approaches and methods" (Pos. 39, 96; 100).

"Successful models for teacher assessment promote educational practice, professional performance, retention of excellent teachers, and performance of underperforming teachers" (Pos. 40; 84; 187; 236).

Teachers' perceptions demonstrate the suggested and most effective form of future evaluation, which requires the use of new elements, as teacher evaluation has become one of the most prominent and controversial topics in education in the 21st century to ensure quality teaching and improve practices, especially in light of distance education [41,46]. Table 3 demonstrates the most essential future evaluation methods according to teachers.

**Table 3.** The most essential future evaluation methods.

| The Most Important Means of Assessment | | Percentage % |
|---|---|---|
| Learning analytics | Portfolio files | 42 |
| | Learners' test results | 55 |
| Classroom observations | The scale of standards for assessing teaching competencies | 44 |
| | Balanced performance scale | 55 |
| | Scales for assessing digital skills | 40 |
| Professional development | Specialized courses and workshops | 36 |
| | Scientific and community activities | 32 |
| | Diligence and professional discipline | 22 |
| Student feedback | Measure learner satisfaction | 20 |
| Peer evaluation | Evaluation of peer teachers | 36 |
| Self-evaluation | Balanced performance scale | 28 |

Teachers must co-design assessment and evaluation systems through collective bargaining [46,50]. Learning data should not be the only or primary source of teacher performance data. Evaluation of teachers can be carried out using comprehensive standards of practice, such as classroom observations, administrator evaluation, teacher–evaluator conferences, portfolios, evidence binders, conference presentations, instructional artifacts that showcase new knowledge and skills [7], formative assessments, peer evaluations,

professional learning communities, other feedback and support, and different metrics of student growth and learning [20,51].

*5.8. Parental Influence on Future Learning*

The study sample indicated the importance of strengthening the connection between the school and the family in improving learning outcomes in the distance education environment, given that following up on learners and controlling them is not only the responsibility of the school but that of the teacher, student, and parents. In this context, instructors identified the following as the most significant roles of parents:

"Parents have a significant role in their children's education through distance learning. They must encourage and guide their children as they navigate the online learning environment" (Pos. 199; 202).

"Parents should monitor their children's progress and communicate regularly with teachers to ensure that their children are meeting academic expectations" (Pos. 76; 199; 202).

"The involvement of parents in distant learning can have a significant impact on their children's achievement and overall educational experience" (Pos. 47; 199; 202).

"Parents must ensure that their children have access to the required resources and technologies" (Pos. 68; 199; 202).

"Parents can contribute to the creation of a conducive learning environment at home by providing a quiet room for studying and minimizing distractions" (Pos. 162; 230; 257).

The interviewed teachers agreed that parents' involvement during the COVID-19 pandemic and school shutdown increased and was affected by the schools' e-learning competence. The study showed that not all schools were ready for e-learning. Schools' suspension of in-person learning prompted parents to search for national curriculum-aligned e-sources. In future distance learning, parents will play a crucial role in their children's education. They need to provide support and guidance to their children in navigating the online learning environment, ensuring that they have access to the necessary resources and technology [52]. Parents should also monitor their children's progress and communicate regularly with teachers to ensure that their children are meeting academic expectations [53]. Additionally, parents can help create a conducive learning environment at home by providing a quiet space for studying and minimizing distractions. Ultimately, parents' involvement in distance learning can greatly impact their children's success and overall educational experience [53,54]. The readiness of teachers, students, and parents is the key to the success of online learning during this emergency; teachers and parents must work well together [32].

## 6. Discussion

The pandemic transition has provided an opportunity to consider how to continue the change in school education to a sustainable paradigm for future generations. In this perspective, the current study sought to comprehend school pupils' experiences in KSA. Careful instructional design, audience consideration, and a systematic implementation model make virtual education effective. So, planning a variety of interconnected elements (e.g., technology, tutorial model, communication, monitoring and follow-up, assessment) produces distinct forms of teaching that must be fitted to the educational objectives. The lessons learned from this research as well as research conducted elsewhere should guide future decisions about implementing online teaching. Key learning axes for future schools can be derived from the following dimensions by comparing the results of the current study with those of previous studies.

*6.1. Future Effectiveness of Teachers*

Future teaching efficacy will be influenced by a number of variables, including technological developments, systemic reforms, and changing student needs. To continue doing their jobs effectively, teachers will need to adjust to new technologies and teaching approaches. Future teachers will need to have a variety of abilities, including digital literacy,

adaptability, collaboration, critical thinking, emotional intelligence, and others. Future learning via distance education will require instructors to fill the following roles: facilitator, mentor, content expert, curriculum designer, assessor, collaborator, technologist, motivator, and advocate. In conclusion, the performance of instructors in the future will depend on their capacity to adjust to shifting conditions and create new talents that address the changing demands of pupils.

### 6.2. Teacher–Student Interaction

The teacher's role is crucial in students' discourses, with their interactions mostly determining student participation in learning. So, encouraging involvement, being available to answer questions and clear up confusion, and showing concern for their learning rhythms and emotional states would minimize this transactional distance, making the instructor seem understanding, adaptable, and close. Virtual environments offer more synchronous and asynchronous communication channels. Chat, forums, music, and emoticons encourage interaction. These communication platforms also favor a younger audience, which puts students closer to dialectic and collaborative learning, where they can even become passive actors, watching others communicate. Rethinking and redesigning relationships between teachers and parents and other caregivers using digital technologies and images as well as words could increase the flow of communication, without exposing teachers to the kind of intrusive and unfair scrutiny of online teaching that sometimes occurred during the pandemic.

### 6.3. Future Educational Technologies

With data-based technologies, educators will be able to create flexible learning spaces and continuous online learning environments, which will spread across the home, schools, and communities. Technology should be integrated as an inherent part of the online teaching and learning process and take into account the pedagogical possibilities associated with online tools, i.e., the suitability and affordances of the various technologies or online resources to make sure that the chosen tools or resources help learners achieve the desired results. Future schools may implement some of the newest trends in educational technology, including big data, artificial intelligence, virtual reality, gamification, mobile learning, collaborative learning, adaptive learning, smart badges, and personalized learning. Overall, these emerging developments in educational technology could transform how schools teach and learn in the future.

### 6.4. Pedagogical Trends

Blended teaching and learning place equal emphasis on what can be accomplished on campus, such as face-to-face engagement, and what can be accomplished online, including offering flexibility and open access to resources and subject matter experts. When greater contact occurs between students, teachers, and outside experts who participate in person or electronically, it is necessary to rethink teaching and learning practices as well as classroom design. In reaction to new technology capabilities, teaching paradigms for both classroom and online delivery must be reevaluated and recalibrated.

Regarding self-directed and non-formal online learning, because of the availability of free open educational materials and social networking, many students will be able to access knowledge without needing to apply for admission in advance, take a predetermined course, or have an instructor. Students may receive help and feedback on their learning from peer discussion and assessment as well as computerized marking. In anywhere, anytime, any size learning, the construction of smaller modules demonstrates the evolution of "any size" learning. Student demand for quick, "just in time" learning modules that meet an instant learning requirement is expanding.

Learning analytics are used to make tracking student learning through digital activities easier and more scalable. Continuous analytical feedback might result in early diagnostics that allow students to focus on areas of weakness prior to the final examination. Instructors

can also utilize analytics to evaluate the quality and usefulness of course materials and track student participation, thereby creating intervention possibilities. Modern accreditation procedures based on competencies increase transparency and acknowledgment of credit transferability and learning.

### 6.5. Resources and Activities

Use of multimedia and open educational resources (OERs): Digital media, open educational resources in the form of short lectures, animations, simulations, virtual labs, virtual worlds, recorded classes, infographics, and many other formats enable instructors and students to access and apply knowledge in a wide variety of ways. Mobile learning, with smartphones, virtual labs, simulators, tablets, and other devices, is provided through online learning. Offering content, quizzes, multimedia resources, and connections among students using mobile devices requires a new look at course design, content packaging, and consideration of the limitations of data packages. How to best integrate mobile devices into course delivery and assessment is a field of continuing exploration. Finally, the link between individual and group learning activities and the integration of resources and activities both relate to the fact that these virtual tools will be created with pedagogical intentions and cross the boundaries between learning and knowledge technologies and information and communication technologies.

### 6.6. New Forms of Learners' Assessment

Digital learning can leave a permanent "trace" in the form of student contributions to online discussion and e-portfolios of work through the collection, storing, and assessment of a student's multimedia online activities. Peer-to-peer forms of assessment promote learners' participation and meaningful engagement. Performance-based assessments, adaptive assessments, peer assessments, game-based assessments, authentic assessments, self-assessments, and social–emotional assessments are a few new types of learner assessments that may be utilized in future schools. Beyond traditional tests or exams, these novel methods of learner assessment can offer a more thorough understanding of a student's abilities. Also, it is possible to gather multisource, heterogeneous, multimodal big data about students' learning processes through artificial intelligence technologies in a smart learning environment, such as the Internet of Things, perception technology, video recording technology, image recognition technology, and platform acquisition technology. With the help of such massive data, it will be possible to better analyze and improve the teaching and learning environments as well as student behavior and performance in the intelligent learning environment.

### 6.7. Teachers' Assessment

Teacher evaluation develops in the context of developed technology. In the digital learning environment, teacher assessment methods include six types. First, student feedback is crucial for evaluating teachers in digital learning contexts. Second, learning analytics: analyzing student learning results with data. Third, peer evaluation: other teachers or field specialists evaluate a teacher's performance. Fourth, self-evaluation: teachers can evaluate themselves by reflecting on their practices and performance. Fifth, classroom observations: administrators or supervisors observe teachers' classroom performance. Sixth, professional development: professional development helps teachers keep up with digital learning trends and best practices.

### 6.8. Smart Infrastructure for Future Schools

Smart infrastructure uses data and technology to improve buildings. These might include energy-efficient lighting and HVAC systems that automatically change based on occupancy and weather conditions and smart sensors that monitor air quality, temperature, and humidity to guarantee a healthy learning environment in schools. High-speed internet throughout the school is used for digital learning and automated building maintenance

and repair systems. Technology and digital scheduling tools may be included. Parents can also access their children's academic progress and school activities online. When building flexible spaces and focusing on sustainability, schools should use renewable energy, reduce waste, promote eco-friendly behaviors among students, improve safety, provide smart access to resources, and foster a happy learning environment.

*6.9. Future Parent Trends*

Meaningful family engagement is the future through stronger family–school engagement and shared priorities. Future family engagement in teaching and learning is unparalleled. Schools may establish learning bubbles where families with children of similar ages may learn together, share resources and technologies, and receive targeted supervision from teachers. Family learning bubbles, where numerous families can remotely learn together, could allow parents to help each other and share supervision tasks and resources. Instructors must help parents manage their kids' work. Schools must provide great home learning materials for parents and students. In future learning, teachers and parents should work together to ensure that students' autonomous learning can be carried out smoothly. Parents not only see whether they learn or not but also the efficiency of learning. They should pay attention to observe whether students can keep up with the teacher's teaching pace and whether they can independently solve learning problems.

The international organizations of education have already pointed out that the way students learn, what they learn, and the skills needed will be radically transformed in the coming years [55]. Smart technologies are ready to come into play, changing the conditions of learning, providing opportunities for transformative learning experiences, and promising more conscious, self-directed, and self-motivated learning [55,56].

According to our findings and previous studies, a technology-enhanced learning environment was beneficial for modifying pedagogical approaches, using fresh teaching techniques, organizing and managing the learning process, and gaining access to relevant information sources. With the advancement of current technology, it is advantageous to establish a smart learning environment where students can increase their intelligence through the use of smart classroom technology and smart pedagogical practices. In addition, technological advancements have enabled students to increase their learning chances, time, and opportunities to enhance their cognition, engagement, and interaction.

All of the foregoing implies that we need to place an emphasis on developing technologies such as big data, AI, learning analytics, mixed reality (MR), meta-verse, and the Internet of Things that are forming a new age for both educators and learners, as well as schools and universities. The application of intelligent technologies such as big data, AI, learning analytics, mixed reality (MR), meta-verse, and the Internet of Things has brought new opportunities and vitality to traditional teaching processes and methods, thereby making the learning process more student-centric. Overall, the authors argue that new smart technologies have the potential to be a crucial development path for future education and a powerful catalyst for education reforms. The implications of the research results highlight the necessity for a new theory that takes into account any new crisis and resilience viewpoint that is based on the COVID-19 pandemic. This entails putting an emphasis on the utilization of modern digital technologies that provide a more interactive experience.

## 7. Conclusions

Lessons learned from COVID-19 can be portrayed as cornerstones of a future paradigm change. To manage the complexity of online education, multimodal approaches to course content objectives can improve learning outcomes. Undaunted, governments must provide dependable communication tools, high-quality digital academic experiences, and technology-enabled learning for students to bridge the gaps in the education system before and after COVID-19, which also required uninterrupted learning. These lofty goals require parents, educators, and community leaders who aspire high, believe sincerely, and demand rigor and results for all children. It will also require allies—thought leaders, trusted institu-

tions, and policymakers—who are willing to support parents, teachers, and school leaders in their educational enterprise renewal.

This study has made evident that some areas of research deserve further attention. Future studies should be carried out to support the hardest hit economically disadvantaged groups. In addition, future research must define the best ways to realize the innovative potential of digital learning technologies while developing clear strategies to manage and mitigate the risks for students. Also, distinctive impacts of online education on students with disabilities should be studied. Teachers and educational stakeholders have to be actively involved in future research designs and discussions. Further research can investigate how to invest the professional capital (that accrues in teaching from human and material resources).

*Limitations*

The analyses conducted on the data in this study did not take into account variables that would have made the participants in this study extremely different, such as the sort of degree they are pursuing, the gender of the teacher, the year they are teaching, the audience they are teaching, etc. Another difficulty was the little time allotted for information collection.

**Author Contributions:** Methodology, U.M.I., D.M.K. and A.Z.A.; Formal analysis, H.M.A., I.M.D. and R.A.A.K.; Investigation, U.M.I., H.M.A. and A.Z.A.; Resources, H.M.A. and D.M.K.; Data curation, R.A.A.K.; Writing—original draft, A.Z.A., I.M.D. and R.A.A.K.; Writing—review and editing, U.M.I.; Project administration, U.M.I.; Funding acquisition, U.M.I. All authors have read and agreed to the published version of the manuscript.

**Funding:** This research has been funded by The Scientific Research Deanship at University of Ha'il-Saudi Arabia through project number (RG-22007).

**Institutional Review Board Statement:** Ll procedures performed in the study were following the ethical standards of the institutional research committee of scientific Research Dean of Hail University (IRB Log Number: RG-22007) and with the 1964 Helsinki declaration and its later amendments.

**Informed Consent Statement:** Informed consent was obtained from the teachers orally.

**Data Availability Statement:** Data is available on request from the authors.

**Conflicts of Interest:** The authors declare no conflict of interest.

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
