# Peer review of "The Drivers of Change for Future Learning: How Teachers Were Taught in the COVID-19 Crisis and What Will Come Next?"

_sustainability, doi:10.3390/su152014766_

Round 1
Reviewer 1 Report
I appreciate the authors; efforts in addressing a significant issue. I am glad to review this manuscript with some comments as follows.
1. COVID-19 Crisis. as a critical theme of this study, deserve a clear and in-depth description regarding threats and opportunities.
2. Some essential concepts are presented in diverse terms, such as online learning, virtual and digital learning, e-learning, distant learning, remote education, online teaching, etc. The authors could unify these terms to enhance the readability and rigor of this research.
3. It's good for the authors to propose several statements with widely cited references. Nevertheless, organized summarization under a specific theoretical base would enhance the academic essence and contributions.
4. Regarding the input of this study, the authors acquire "written responses" from participants. Without proper inquiry and interaction, it might suffer from getting only superficial opinions.
5. The readers would be interested in more details about the application of MAXQDA.
6. For the significant findings presented in Table 2, there seem to be inconsistencies in the number of open and axial codes. Higher-order summarization and categorization are preferred to capture the essential findings.
7. Strong links between the findings and discussion are expected.
8. The authors could discuss and stress this study's theoretical and managerial implications.
9. The limitation of "the little time allotted for information collection" seems not a good reason. The authors could also suggest directions for future studies.
Author Response
|
Reviewer 1 |
|
|
1. COVID-19 Crisis. as a critical theme of this study, deserve a clear and in-depth description regarding threats and opportunities. |
The COVID-19 Crisis was dealt with some specialization by identifying its educational effects under the title (Educational crises effect) we think this is enough. |
|
2. Some essential concepts are presented in diverse terms, such as online learning, virtual and digital learning, e-learning, distant learning, remote education, online teaching, etc. The authors could unify these terms to enhance the readability and rigor of this research. |
Very good note. We try to do it. |
|
3. It's good for the authors to propose several statements with widely cited references. Nevertheless, organized summarization under a specific theoretical base would enhance the academic essence and contributions. |
Very good note.. the research team will try to take it |
|
4. Regarding the input of this study, the authors acquire "written responses" from participants. Without proper inquiry and interaction, it might suffer from getting only superficial opinions. |
Through the analysis and study of the data, the research team excluded any superficial opinions before coding them |
|
5. The readers would be interested in more details about the application of MAXQDA. |
Leading qualitative data analysis software MAXQDA has become popular recently. The MAXQDA program performs many qualitative analysis and research tasks, including content analysis, data classification, and binary analysis, by focusing on the relationships between elements, cross-classifying and analyzing data, linking data, and searching for patterns and concepts. Textual analysis and thematic analysis, which involves comprehending texts' semantics and meanings and finding patterns (Marjaei et al., 2019, Gizzi & Rädiker, 2021). |
|
7. Strong links between the findings and discussion are expected. |
Work was done to find immediate links between the results and the discussion |
|
8. The authors could discuss and stress this study's theoretical and managerial implications. |
Done as we can |
|
9. The limitation of "the little time allotted for information collection" seems not a good reason. The authors could also suggest directions for future studies. |
We add some suggested directions for future studies ((Future Studies should be carried out to support the hardest hit economically disadvantaged groups. In addition, future research must define the best ways to realize the innovative potential of digital learning technologies while developing clear strategies to manage and mitigate the risks for students)) |

Reviewer 2 Report
The article is well structured and designed, clearly reporting reseach questions methodology and results. Results are coherent with research questions and theoretical framework and precisely described and detailed.
It would be better to name the opprtunistic or emergent samplig a technique or method of research rather than a methodology.
Furthermore, there are some typos:
Paragraph 3.4, 5th bullet point: . predicts;
Paragraph Findings, 5th line: ; The.
In conclusion, the bibliography needs to be put in alphabetical order and the formatting needs to be aligned (some author’s name are in bold others not; some references are in light blue and other in black).
Author Response
|
comment |
action |
|
Reviewer 2 |
|
|
The article is well structured and designed, clearly reporting research questions, methodology, and results. Results are coherent with research questions and theoretical framework and precisely described and detailed. |
All Thanks for your opinion. |
|
It would be better to name opportunistic or emergent sampling a technique or method of research rather than a methodology. |
Done |
|
Furthermore, there are some typos: Paragraph 3.4, 5th bullet point: predicts; Paragraph Findings, 5th line; The. |
Done
|
|
In conclusion, the bibliography needs to be put in alphabetical order and the formatting needs to be aligned (some authors’ names are in bold, others not; some references are in light blue and others in black). |
Done |

Reviewer 3 Report
The manuscript entitled “The Drivers of Change for Future Learning: How Teachers were Taught in the COVID-19 Crisis and What Will Come Next?” analyzed the learning and studying trend in the COVID. I am glad that this manuscript has a promising perspective for existing literature, and its research findings may guide future decisions about implementing online teaching, and the dimensions identified in this research can be compared with previous studies to derive key learning axes for future schools. Below shows the reviewer’s major concerns and hope the authors may respond accordingly.
1. The first research question of this study is very inclusive. For example, perceptions may from a large list of perspectives such as students’ learning behavior changes, learning environment challenges, or even students’ psychological traits. So I recommend using a smaller scope to describe this research question. For example, you can see what are answered in this study, and then revise your research question.
2. The study suggests several areas for future research to improve the structure of e-learning, infrastructure, and technological equipment in schools. However, some additional perspectives may improve this study. For example, this study mainly included qualitative data from interviews with teachers. So a quantitative analysis that includes data on student learning outcomes may provide more insights into the effectiveness of online teaching.
3. I am still a little bit confused on the data. The data information could be improved in the revision. For example, the authors claimed that “The open interview comprises a set of pre-planned, consecutive questions that are asked to all participants in the same format and system.”. Then, what are those questions asking about? Maybe one example or a describable summary could be helpful to address this part.
4. What is MAXQDA and what is is used for in this study? A interpretation to MAXQDA will help readers not familiar with this qualitative analysis tool.
5. I am glad to find and agree that the key dimensions identified in this research for implementing online teaching are careful instructional design, audience consideration, and a systematic implementation model that produces various forms of teaching tailored to educational objectives. Therefore, tailored learning should be an important perspective and focus discussed in the literature review section. However, I didn’t find enough discussion in the literature review. When you discussed technology by discussing the data-driven learning, I highly recommend you could expand a little bit by giving some examples. For example, there is an increasing use of statistical topic model to predict students' academic performance and promote tailored learning by providing individual learning recommendation such as
a. Xiong, J., Wheeler, J. M., Choi, H. J., & Cohen, A. S. (2022). A Bi-level Individualized Adaptive Learning Recommendation System Based on Topic Modeling. In Quantitative Psychology: The 86th Annual Meeting of the Psychometric Society, Virtual, 2021 (pp. 121-140). Cham: Springer International Publishing.
b. Apaza, R. G., Cervantes, E. V., Quispe, L. C., & Luna, J. O. (2014). Online Courses Recommendation based on LDA. In SIMBig (pp. 42-48).
There is also research discussing the use of assessment to evaluate learners’ learning efficiency, skill and knowledge, such as
a. Baylari, A., & Montazer, G. A. (2009). Design a personalized e-learning system based on item response theory and artificial neural network approach. Expert Systems with Applications, 36(4), 8013-8021.
I believe the introduction of these technologies can make the conclusion of your manuscript solid since these technologies are playing an important role in tailored and individualized learning in educational research. You may also consult a recently published Sustainability article for what they addressed the literature on tailored online learning: Zhang, J., Qiu, F., Wu, W., Wang, J., Li, R., Guan, M., & Huang, J. (2023). E-Learning Behavior Categories and Influencing Factors of STEM Courses: A Case Study of the Open University Learning Analysis Dataset (OULAD). Sustainability, 15(10), 8235.
Overall, this is a good manuscript with promising research objectives for future learning analyses. I am looking forward to reading the revision of this manuscript.
The English of this manuscript is okay, and minor proofreading work is required in the revision.
Author Response
|
Reviewer 3 |
|
|
The first research question of this study is very inclusive. For example, perceptions may from a large list of perspectives, such as students’ learning behavior changes, learning environment challenges, or even students’ psychological traits. So I recommend using a smaller scope to describe this research question. For example, you can see what are answered in this study and then revise your research question. |
Thank you for your note It is very difficult to change the question at this stage. |
|
The study suggests several areas for future research to improve the structure of e-learning, infrastructure, and technological equipment in schools. However, some additional perspectives may improve this study. For example, this study mainly included qualitative data from interviews with teachers. So a quantitative analysis that includes data on student learning outcomes may provide more insights into the effectiveness of online teaching. |
Good note |
|
What is MAXQDA, and what is used for in this study? An interpretation to MAXQDA will help readers not familiar with this qualitative analysis tool |
We add an explanation as follow:
Leading qualitative data analysis software MAXQDA has become popular recently. The MAXQDA program performs many qualitative analysis and research tasks, including content analysis, data classification, and binary analysis, by focusing on the relationships between elements, cross-classifying and analyzing data, linking data, and searching for patterns and concepts. Textual analysis and thematic analysis, which involves comprehending texts' semantics and meanings and finding patterns (Marjaei et al., 2019, Gizzi & Rädiker, 2021). |
|
I am glad to find and agree that the key dimensions identified in this research for implementing online teaching are careful instructional design, audience consideration, and a systematic implementation model that produces various forms of teaching tailored to educational objectives. |
Thanks for your opinion |
|
performance and promote tailored learning by providing individual learning recommendations such as - Xiong, J., Wheeler, J. M., Choi, H. J., & Cohen, A. S. (2022). A Bi-level Individualized Adaptive Learning Recommendation System Based on Topic Modeling. In Quantitative Psychology: The 86th Annual Meeting of the Psychometric Society, Virtual, 2021 (pp. 121-140). Cham: Springer International Publishing.
- Apaza, R. G., Cervantes, E. V., Quispe, L. C., & Luna, J. O. (2014). Online Courses Recommendation based on LDA. In SIMBig (pp. 42-48).
There is also research discussing the use of assessment to evaluate learners’ learning efficiency, skill, and knowledge, such as:
Baylari, A., & Montazer, G. A. (2009). Design a personalized e-learning system based on item response theory and artificial neural network approach. Expert Systems with Applications, 36(4), 8013-8021. |
Done, we use this citation. It is good reference.
Xiong, J., Wheeler, J. M., Choi, H. J., & Cohen, A. S. (2022). A Bi-level Individualized Adaptive Learning Recommendation System Based on Topic Modeling. In Quantitative Psychology: The 86th Annual Meeting of the Psychometric Society, Virtual, 2021 (pp. 121-140). Cham: Springer International Publishing |
|
I believe the introduction of these technologies can make the conclusion of your manuscript solid since these technologies are playing an important role in tailored and individualized learning in educational research. You may also consult a recently published Sustainability article for what they addressed the literature on tailored online learning: Zhang, J., Qiu, F., Wu, W., Wang, J., Li, R., Guan, M., & Huang, J. (2023). E-Learning Behavior Categories and Influencing Factors of STEM Courses: A Case Study of the Open University Learning Analysis Dataset (OULAD). Sustainability, 15(10), 8235. |
Done
Zhang, J., Qiu, F., Wu, W., Wang, J., Li, R., Guan, M., & Huang, J. (2023). E-Learning Behavior Categories and Influencing Factors of STEM Courses: A Case Study of the Open University Learning Analysis Dataset (OULAD). Sustainability, 15(10), 8235. |

Round 2
Reviewer 1 Report
I appreciate the authors' efforts in revising their manuscript with significant improvements. Still, I propose some suggestions as follows.
1. It is strongly suggested to have a solid theoretical framework to compare research findings.
2. The authors could summarize the findings from applying MAXQDA with broader concepts.
3. The authors could comment more in-depth on their findings' theoretical and managerial implications.
4. Some possible typos and grammar errors could be double-checked and fixed.
Author Response
|
Comment |
Action |
|
Reviewer |
|
|
The authors could summarize the findings by applying MAXQDA with broader concepts. |
Leading qualitative data analysis software MAXQDA has become popular recently. The MAXQDA program performs many qualitative analysis and research tasks, including content analysis, data classification, and binary analysis, by focusing on the relationships between elements, cross-classifying and analyzing data, linking data, and searching for patterns and concepts. Textual analysis and thematic analysis, which involves comprehending texts' semantics and meanings and finding patterns (Marjaei et al., 2019, Gizzi & Rädiker, 2021). |
|
The authors could summarize the findings by applying MAXQDA with broader concepts. |
We did our best |
|
It is strongly suggested to have a solid theoretical framework to compare research findings. |
All thanks for your suggestion, we do our best to enrich the article, If you suggest exact changes tell us |
Reviewer 3 Report
The revision looks good to me. I am happy to see the improvement of this work.
The English level of this work is readable and no obvious issues were identified.
Author Response
Dear
All thanks for your Additional commants that help me to improvment our manuscript.
Round 3
Reviewer 1 Report
I appreciate the authors' significant enhancements in revising their manuscript. My primary suggestion lies in "6. Discussion," after itemized discussions, it would be more insightful for the authors to propose a broader summarization and comments on their research findings.
Author Response
Dear Reviewer,
I hope you are doing well and have a safe.
Thanks a lot for your valuable comments; we will do our best to consider all of them.
As your comment ((it would be more insightful for the authors to propose a broader summarization and comments on their research findings)), We added this paragraph before the conclusion:
The international organizations of education have already pointed out that the way students learn, what they learn, and the skills needed, will be radically transformed in the coming years (Drigas, et al., 2023). Smart technologies are ready to come into play, changing the conditions of learning, providing opportunities for transformative learning experiences, and promising more conscious, self-directed and self-motivated learning (Drigas, et al., 2023).
According to our findings and those of previous studies, a technology-enhanced learning environment was beneficial for modifying pedagogical approaches, using fresh teaching techniques, organizing and managing the learning process, and gaining access to relevant information sources. With the advancement of current technology, it is advantageous to establish a smart learning environment where students can increase their intelligence through the use of smart classroom technology and smart pedagogical practices. In addition, technological advancements enabled students to increase their learning chances, time, and opportunities to enhance their cognition, engagement, and interaction.
All of the foregoing implies that we need to place an emphasis on developing technologies such as Big data, AI, Learning analytics, Mixed reality (MR), meta-verse, and the internet of things that are forming a new age for both educators and learners, as well as schools and universities. The application of intelligent technologies such as Big data, AI, Learning analytics, Mixed reality (MR), meta-verse, and the internet of things has brought new opportunities and vitality to traditional teaching processes and methods, thereby making the learning process more student-centric. Overall, the authors argue that new smart technologies have the potential to be a crucial development path for future education and a powerful catalyst for education reforms. The implications of the research results highlight the necessity for a new theory that takes into account any new crisis and resilience viewpoint that is based on the COVID-19 pandemic. This entails putting an emphasis on the utilization of modern digital technologies that provide a more interactive experience.
We hope it adds more value to our article.
All Regards,
Prof. Usama Ibrahem
